



# Atmospheric propane (C$_3$H$_8$) column retrievals from ground-based FTIR observations at Xianghe, China

Minqiang Zhou[1], Pucai Wang[1], Bart Dils[2], Bavo Langerock[2], Geoff Toon[3], Christian Hermans[2], Weidong Nan[4], Qun Cheng[4], and Martine De Mazière[2]

[1]Institute of Atmospheric Physics, Chinese Academy of Sciences, Beijing, China
[2]Royal Belgian Institute for Space Aeronomy (BIRA-IASB), Brussels, Belgium
[3]Jet Propulsion Laboratory, California Institute of Technology, Pasadena, CA, USA
[4]Xianghe Observatory of Whole Atmosphere, Institute of Atmospheric Physics, Chinese Academy of Sciences, Xianghe, China

**Correspondence:** Minqiang Zhou (minqiang.zhou@mail.iap.ac.cn); Pucai Wang (pcwang@mail.iap.ac.cn)

**Abstract.** Propane (C$_3$H$_8$) is an important trace gas in the atmosphere, as it is a proxy for oil and gas production and has a significant impact on atmospheric chemical reactions related to the hydroxyl radical and tropospheric ozone formation. In this study, solar direct absorption spectra near 2967 cm$^{-1}$ recorded by a ground-based Fourier Transform InfraRed spectrometer (FTIR) are applied to retrieve C$_3$H$_8$ total columns between June 2018 and July 2022 at Xianghe in North China. The systematic

and random uncertainties of the C$_3$H$_8$ column retrieval are estimated to be 18.2% and 18.1%, respectively. The mean and standard deviation of the C$_3$H$_8$ columns derived from the FTIR spectra at Xianghe are $1.80\pm0.81(1\sigma) \times 10^{15} \ molecules/cm^2$. Good correlations are found between C$_3$H$_8$ and other non-methane hydrocarbons, such as C$_2$H$_6$ (R=0.84) and C$_2$H$_2$ (R=0.79), as well as between C$_3$H$_8$ and CO (R=0.72). However, the correlation between C$_3$H$_8$ and CH$_4$ is relatively weak (R=0.45). The FTIR C$_3$H$_8$ measurements are also compared against two atmospheric chemical transport model simulations (the Whole

Atmosphere Community Climate Model (WACCM) and the Copernicus Atmosphere Monitoring Service (CAMS)). We find that the C$_3$H$_8$ columns from both models have different seasonal variations as compared to the FTIR measurements. Moreover, the mean C$_3$H$_8$ columns derived from the WACCM and CAMS models are about 68% larger than the FTIR retrievals. The new FTIR measurements at Xianghe provide us an insight into the C$_3$H$_8$ column variations and underlying processes in North China.

## 15  1  Introduction

Methane (CH$_4$) and non-methane hydrocarbons (NMHC), such as ethane (C$_2$H$_6$), acetylene (C$_2$H$_2$), propane (C$_3$H$_8$), propene (C$_3$H$_6$), and isoprene (C$_5$H$_8$), are important trace gases that play significant roles in atmospheric chemical reactions related to hydroxyl radical (OH) abundance and tropospheric ozone (O$_3$) formation (Sze, 1977; Donahue and Prinn, 1990; Tan et al., 2012; Lelieveld et al., 2015). Human activities contribute greatly to the emissions of CH$_4$ and NMHCs, especially in urban

areas (Bourtsoukidis et al., 2019; Saunois et al., 2020). Atmospheric C$_2$H$_6$ and C$_3$H$_8$ emissions are dominated by oil and



gas sources, and they are co-emitted with $CH_4$. Therefore, numerous studies used the ratio of $C_2H_6$ and/or $C_3H_8$ to $CH_4$ to understand the $CH_4$ trend (Kort et al., 2016; Franco et al., 2016; Rigby et al., 2017).

The major sink of $C_2H_6$ and $C_3H_8$ is the reaction with OH, and the lifetime of $C_3H_8$ and $C_2H_6$ is about 2-4 weeks in summer and 2 months in winter (Jacob, 1999; Xiao et al., 2008). Compared to $CH_4$ with a lifetime of the order of 10 years (IPCC, 2013),

the short-lived gases $C_2H_6$ and $C_3H_8$ are not well-mixed on the global scale, and are therefore more representative of regional pollution as is carbon monoxide (CO) (Toon et al., 2021).

Atmospheric $C_3H_8$ concentrations at the surface are observed by National Oceanic and Atmospheric Administration (NOAA) - Global Monitoring Laboratory (GML) flask sampling measurements at 12 sites (https://gml.noaa.gov/hats/gases/C3H8.html). In addition, the HIAPER Pole-to-Pole Observations (HIPPO), Atmospheric Tomography (ATom), and In-service Aircraft for a

10 Global Observing System (IAGOS) aircraft campaigns provide in-situ gas analyzer measurements of $C_3H_8$ with a wide latitudinal coverage, particularly in the Pacific Ocean, Atlantic Ocean, Europe and North America (Wofsy, 2011; Thompson et al., 2022; Li et al., 2022). Toon et al. (2021) has demonstrated the use of $C_3H_8$ absorption lines in the mid-infrared region (Harrison et al., 2010), in solar absorption spectra from MkIV interferometers for retrieving the $C_3H_8$ total columns or vertical profiles at several locations in Sweden, the USA, and Antarctica. Solar absorption infrared spectra are also being collected by ground-

15 based Fourier Transform Spectrometers (FTIR) within the Network for the Detection of Atmospheric Composition Change - InfraRed Working Group (NDACC-IRWG) (De Mazière et al., 2018). Currently, there are more than 20 NDACC-IRWG global sites, with a good global latitudinal coverage from 78°S to 80°N (https://www2.acom.ucar.edu/irwg/sites). However, to our knowledge, no site has reported $C_3H_8$ retrievals from spectra observed by a Bruker 125HR spectrometer within the NDACC-IRWG.

Xianghe (39.75 °N, 116.96 °E) is located in North China, about 50 km east of the mega-city Beijing (Yang et al., 2020). According to the Emissions Database for Global Atmospheric Research (EDGAR) v6.0 (Crippa et al., 2020) and the Multi-resolution Emission Inventory for China (MEIC) inventory (Wang et al., 2015; Li et al., 2017), there is a large $CH_4$ emission source in North China coming from fuel exploitation and oil refineries. Therefore, we expect that the $C_2H_6$ and $C_3H_8$ concentrations are relatively high in this region. In June 2018, a Bruker IFS 125HR spectrometer, compliant with the NDACC-IRWG

protocol, started recording solar absorption spectra in the mid-infrared spectral range. The spectra have been used to retrieve several atmospheric components, e.g., $O_3$, $CH_4$, CO, $C_2H_2$, $C_2H_6$, HCN and $H_2CO$ (Ji et al., 2020; Zhou et al., 2020, 2021, 2023; Vigouroux et al., 2020; Sha et al., 2021). In this study, we investigate the $C_3H_8$ retrieval from ground-based FTIR spectra at Xianghe, and discuss the $C_3H_8$ column variation in North China, based on these new FTIR measurements.

The remainder of this paper is organized as follows. Section 2 describes the Xianghe FTIR site and $C_3H_8$ retrieval method,

Section 3 presents the $C_3H_8$ variations and correlations with other species. Moreover, the $C_3H_8$ measurements at Xianghe are compared to model simulations and ground-based MkIV measurements at other places. Finally, Section 4 draws a conclusion.



## 2 Method

### 2.1 Xianghe FTIR spectra measurement

The Xianghe FTIR measurement system started in June 2018, and has been well described in previous studies (Yang et al., 2020; Zhou et al., 2021, 2023). Briefly, the FTIR measurement system contains 3 parts: a solar tracker system, a weather station, and a Bruker IFS 125HR Fourier-transform infrared (FTIR) spectrometer. Short-Wave infrared (SWIR) and Near-infrared (NIR) spectra (4000-11000 $cm^{-1}$) with a spectral resolution of 0.02 $cm^{-1}$ are recorded with an InGaAs detector, and these spectra are used to derive greenhouse gases total column abundances as a contribution to the Total Carbon Column Observing Network (TCCON). Mid-infrared (MIR) spectra (1800-4500 $cm^{-1}$), with a spectral resolution of 0.0035-0.0070 $cm^{-1}$, are recorded with an InSb detector. To enhance the signal-to-noise ratio (SNR) of the spectra, we add specific optical filters into the light path when recording each MIR spectrum as recommended by NDACC-IRWG (Blumenstock et al., 2021; Zhou et al., 2023). A typical MIR spectrum used for $C_3H_8$ retrieval is shown in Figure 1. Note that, we only operate the FTIR measurement during the daytime and under clear-sky conditions, as the sun is the light source. In general, we carry out 4 to 10 MIR spectral measurements of this type per day for about 200 days per year. The spectra taken between June 2018 and July 2022 (about 4 years) are used in this study.

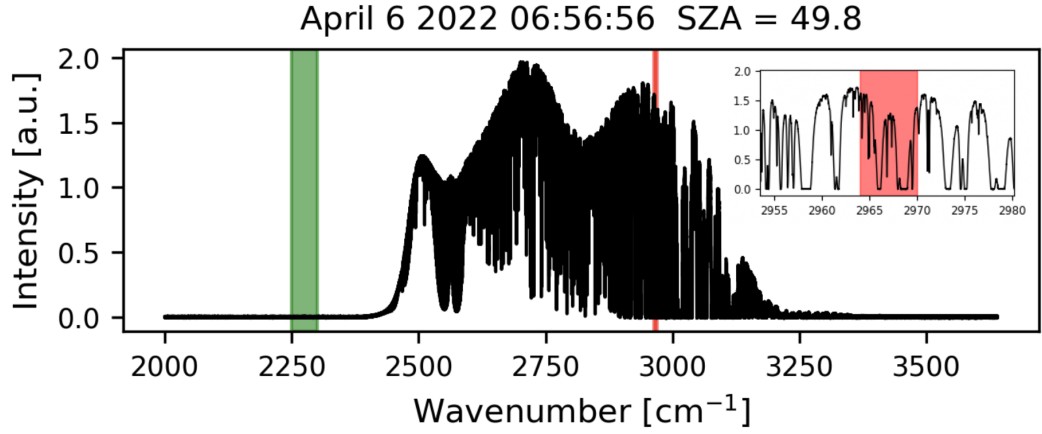

**Figure 1.** A typical MIR spectrum observed at Xianghe on 6 April 2022 with a solar zenith angle of 49.8°. The red and green windows indicate the micro-windows windows used for the $C_3H_8$ retrieval and for calculating the noise (Eq. 5), respectively. The insert in the right-hand corner shows a zoom on the retrieval micro-window.

### 2.2 Retrieval method

To derive $C_3H_8$ mole fractions from the observed spectra, we follow the optimal estimation methodology (Rodgers, 2000). The forward model ($\mathbf{F}$) simulates the absorption spectra ($\boldsymbol{y}$) observed by the FTIR system. It includes modelling of the solar spectra at the top of the atmosphere (TOA), the physics of the radiative transfer from the TOA to the ground-based FTIR, and


the FTIR spectrometer line shape function (ILS). Then, the observed spectra ($\boldsymbol{y}$) can be written as

$$\boldsymbol{y} = \boldsymbol{F}(\boldsymbol{x}, \boldsymbol{b}) + \boldsymbol{\epsilon}, \tag{1}$$

where $\boldsymbol{x}$ is the state vector (retrieved parameters), $\boldsymbol{b}$ is the forward model parameters (not retrieved), and $\boldsymbol{\epsilon}$ is the error, including the measurement noise and forward model errors. We wish to find the optimal state ($\boldsymbol{x}$) that minimize the cost function ($\boldsymbol{J}(\boldsymbol{x})$), given by

$$\boldsymbol{J}(\boldsymbol{x}) = [\boldsymbol{y} - \boldsymbol{F}(\boldsymbol{x})]^T \mathbf{S}_{\boldsymbol{\epsilon}}^{-1} [\boldsymbol{y} - \boldsymbol{F}(\boldsymbol{x})] + [\boldsymbol{x} - \boldsymbol{x_a}]^T \mathbf{S}_R [\boldsymbol{x} - \boldsymbol{x_a}], \tag{2}$$

where $\mathbf{S}_{\boldsymbol{\epsilon}}$ is the measurement error covariance matrix; $\mathbf{S}_R$ is the regularization matrix; $\boldsymbol{x_a}$ is the *a priori* state vector. The Levenberg-Marquardt (LM) method is used to iteratively solve the above equation:

$$\boldsymbol{x}_{i+1} = \boldsymbol{x}_i + [(1+\gamma)\mathbf{S}_R + \mathbf{K}_i^T \mathbf{S}_{\boldsymbol{\epsilon}}^{-1} \mathbf{K}_i]^{-1} \left\{ \mathbf{K}_i^T \mathbf{S}_{\boldsymbol{\epsilon}}^{-1} [\boldsymbol{y} - \mathbf{F}(\boldsymbol{x_i})] - \mathbf{S}_R [\boldsymbol{x}_i - \boldsymbol{x}_a] \right\}, \tag{3}$$

where $\mathbf{K}$ is the Jacobian matrix, $\gamma$ is a parameter to adjust the regularization of a priori information in each iteration step (Rodgers, 2000). Upon convergence, the final state is called $\boldsymbol{x}_r$, which can be related to the true state ($\boldsymbol{x}_t$):

$$\boldsymbol{x}_r = \boldsymbol{x}_a + \mathbf{A}(\boldsymbol{x}_t - \boldsymbol{x}_a) + \varepsilon, \tag{4}$$

where $\mathbf{A}$ is the averaging kernel matrix, representing the sensitivity of the retrieved parameters to the true parameter, and $\varepsilon$ is the retrieval uncertainty propagated from Eq.1.

## 2.3 Retrieval strategy

In this study, we use the SFIT4 v1.0 retrieval algorithm (Pougatchev et al., 1995; Hase et al., 2004) to perform the forward model simulation as well as the LM inversion. The well-established SFIT4 code has been used extensively to retrieve total/partial column of atmospheric species in the NDACC-IRWG community (Zhou et al., 2016; De Mazière et al., 2018; Ortega et al., 2019).

The key $C_3H_8$ retrieval parameters used in this study are listed in Table 1. The retrieval window is set to 2964.5-2970.0 cm$^{-1}$, where we have the strongest $C_3H_8$ absorption line (Harrison et al., 2010). Apart from $C_3H_8$, several interfering gases ($H_2O$, $CH_4$, $O_3$, and HDO) also have absorption lines in this window as shown in Figure 2. To reduce the impact of uncertainties about the abundances of these species, these column abundances are retrieved along with the target gas mole fractions; only for $H_2O$ we perform a profile retrieval, because of its large variability.

The chosen spectroscopic parameters are crucial in the remote sensing technique. In this study, we have tested several line lists, particularly for $H_2O$ (HDO) and $CH_4$ (see Table 2), including DLR2016 (Loos et al., 2017), HITRAN2020 (Gordon et al., 2022) and ATM2020 (https://mark4sun.jpl.nasa.gov/pseudo.html). The ATM2020 line list is created by Geoff Toon (NASA, JPL) based on HITRAN2020 together with some additional atmospheric and laboratory measurements. It includes pseudo linelists (PLL) for certain species as the ones we use for $C_3H_8$, based on laboratory cross section measurements by Harrison et al. (2010). For $C_2H_6$, we use HITRAN2020. We tested more than 1000 spectra recorded in 2019 at Xianghe, and we observed





that the lowest root-mean-square error (RMSE) of the fitting residual is obtained when the ATM2020 spectral database is used for $CH_4$ and $H_2O$. Table 1 lists the spectral datasets finally used for each species in the $C_3H_8$ retrieval strategy.

**Table 1.** The retrieval window, interfering specie, spectroscopy, fitting parameters for $C_3H_8$ at Xianghe.

| Parameters | settings |
|---|---|
| Retrieval window ($cm^{-1}$) | 2964.5-2970.0 |
| Profile retrieval species | $C_3H_8$, $H_2O$ |
| Column retrieval species | $C_2H_6$, $CH_4$, HDO |
| Retrieved parameters | slope, phase, instrument line shape, wavenumber shift |
| | solar intensity, solar wavenumber shift |
| A priori profile | NCEP for $H_2O$, HDO; WACCM for $C_2H_6$, $C_3H_8$, $CH_4$ |
| Spectroscopy | PLL for $C_3H_8$; ATM20 for $H_2O$, HDO, $CH_4$; HITRAN2020 for $C_2H_6$ |
| Regularization | Tikhonov $\mathbf{L}_1$ method |
| DOFS | 1.1 |

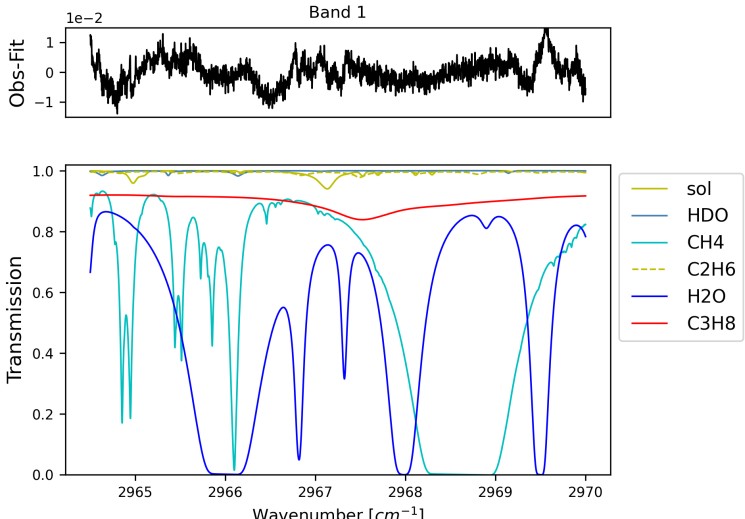

**Figure 2.** The transmittances of main species and solar lines (bottom), as well as the fitting residual (top) using the typical spectrum shown in Figure 1 in this retrieval window.

The *a priori* profiles for $C_3H_8$, $C_2H_6$, and $CH_4$ are derived from the Whole Atmosphere Community Climate Model (WACCM) version 6. We use the averages of the monthly means between 1980 and 2040 (61 years) as the *a priori* profiles. Since the variations of temperature and humidity are quite large in the atmosphere, using fixed *a priori* profiles often





**Table 2.** The fitting RMSE of the retrieval window for all spectra in 2019 from several different line lists.

| $H_2O$ (HDO) | $CH_4$ | RMSE (mean$\pm 1\sigma$) |
|---|---|---|
| ATM2020 | ATM2020 | $0.925\pm0.241$ |
| HITRAN2020 | ATM2020 | $0.967\pm0.267$ |
| DLR | ATM2020 | $0.968\pm0.267$ |
| ATM2020 | HITRAN2020 | $1.233\pm0.281$ |
| HITRAN2020 | HITRAN2020 | $1.314\pm0.289$ |

results in a bad fitting, especially for the first iteration. To provide a better estimation of temperature and humidity profiles, for each measurement, the $H_2O$ (HDO) and temperature vertical profiles are derived from the closest 6-hourly NCEP reanalysis data (Saha et al., 2014), and linearly interpolated to the measurement time.

According to Eq.2, the cost function $J(x)$ is composed of the measurement and *a priori* information, each contracted with a weight matrix $\mathbf{S}_\epsilon$ and $\mathbf{S}_R$, respectively. In this study, the diagonal of the $\mathbf{S}_\epsilon$ is calculated as $1/\text{SNR}^2$, and the non-diagonal values are set to 0. The SNR is calculated as

$$\text{SNR} = \frac{\overline{I_r}}{\sigma_{I_n}}, \tag{5}$$

where $\overline{I_r}$ is the max radiation intensity in the $C_3H_8$ retrieval window (2964.5-2970.0 cm$^{-1}$; red window in Figure 1) and $\sigma_{I_n}$ is the standard deviation (std) of the intensity in the noise window (2250.0-2300.0 cm$^{-1}$; green window in Figure 1). The Tikhonov $\mathbf{L}_1$ regularization method (Tikhonov, 1963) is applied to generate the $\mathbf{S}_R$, with

$$\mathbf{S}_R = \alpha \mathbf{L}_1^T \mathbf{L}_1, \tag{6}$$

$$L_1 = \begin{bmatrix} -1 & 1 & 0 & \dots & 0 & 0 \\ 0 & -1 & 1 & \dots & 0 & 0 \\ \vdots & \vdots & \vdots & \ddots & \vdots & \vdots \\ 0 & 0 & 0 & \dots & -1 & 1 \end{bmatrix}. \tag{7}$$

To determine the $\alpha$ value in Eq. 6, we apply the degree of freedom for signal (DOF) method proposed by Steck (2002). The trace of the averaging kernel matrix ($\mathbf{A}$) is the DOF, indicating the pieces of independent information of the retrieval (Rodgers, 2000). First, we use the optimal estimation method (OEM) to get an estimated DOF. $\mathbf{S}_R$ using the OEM is derived from a covariance matrix on the WACCM monthly means between 1980 and 2040 ($S_{R i,i} = S_{i,i}^{-1} = \sigma_i^{-2}$; diagonal values), and the non-diagonal values are set as

$$S_{R i,j}^{-1} = S_i j = e^{(d_{i,j}/4)}/(\sigma_i \sigma_j), \tag{8}$$

where $d_{i,j}$ is the vertical distance between layer $i$ and layer $j$, in km. The DOF derived from the OEM is about 1.1, indicating that there is only column information for the $C_3H_8$ retrieval. Knowing that, we tune the $\alpha$ value in Eq.6 to make the DOF derived from the Tikhonov method close to the DOF that is derived from the OEM; this approach results in setting $\alpha$ to 1000.





## 2.4 Retrieval uncertainty

The retrieval error ($\epsilon$) of the FTIR $C_3H_8$ column contains three parts as

$$(\mathbf{A} - \mathbf{I})(\boldsymbol{x}_t - \boldsymbol{x}_a) \quad ... \quad smoothing\ error \tag{9}$$

$$\mathbf{G}_y\mathbf{K}_b(\boldsymbol{b}_t - \boldsymbol{b}_a) \quad ... \quad model\ parameter\ error \tag{10}$$

$$\mathbf{G}_y\epsilon \quad ... \quad measurement\ error \tag{11}$$

where $\mathbf{G}_y$ is the contribution function; $\boldsymbol{b}_t$ and $\boldsymbol{b}$ are the true and used model inputs, respectively. Table 3 lists the systematic and random uncertainty of each component. For the smoothing error, we separate the contributions into target species ($C_3H_8$), interfering species ($H_2O$, HDO, $CH_4$, $C_2H_6$), and retrieved parameters (slope, phase, wavenumber shift, instrument line shape, solar intensity and shift). For the model parameter contributions, we calculate the $C_3H_8$ uncertainty contribution coming from

spectroscopy, solar zenith angle (SZA), temperature profile, curvature parameter, and zero level shift (zshift). The systematic and random uncertainties of each parameter are also listed in Table 3. The vertical distributions of the systematic and random uncertainties are shown in Figure 3. Note that the spectroscopy uncertainty in Table 3 is the sum of the uncertainties from the line intensity, pressure-dependent parameter (linePAir) and temperature-dependent parameter (lineTAir). Based on our uncertainty estimation, the total systematic and random uncertainty of the $C_3H_8$ column are both about 18%, and the dominating

contribution is the uncertainty on the background curvature parameter in the forward model. To represent the variability of the $C_3H_8$, we select all days with at least 3 individual measurements on each day, and calculate the daily std. The average of all the daily stds is about 15.3%, and it is close to our estimated random uncertainty.

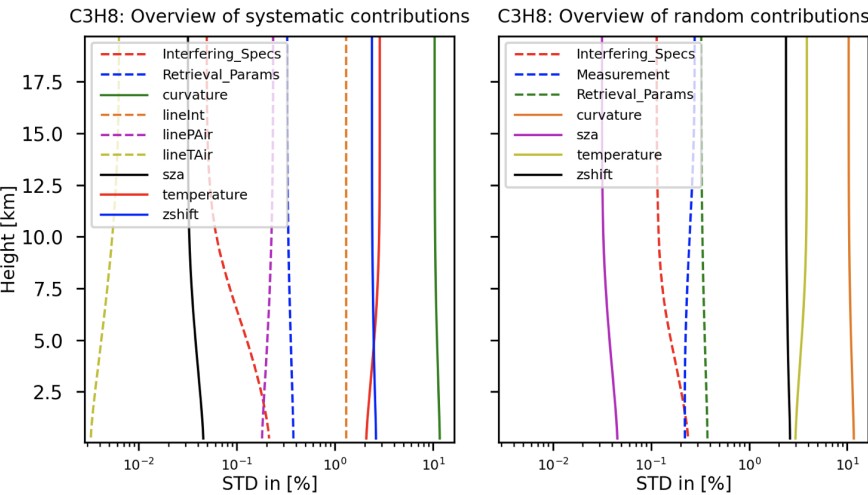

**Figure 3.** The vertical profiles of the systematic (left) and random error (right) of the FTIR $C_3H_8$ retrieval from each component.





**Table 3.** The systematic and random (sys/ran) retrieval uncertainties for the total columns of $C_3H_8$. The '-' means that the uncertainty is less than 0.1%. 1 $\sigma$ of the target or interfering species is the std derived from the WACCM model monthly means between 1980 and 2040. The relative std in the bottom row is the average of daily std of $C_3H_8$ columns on all days with at least 3 measurements, which is to represent the variability of the retrieval.

| Error source | Parameter | Parameter uncertainty (sys/ran) | $C_3H_8$ column uncertainty [%] |
|---|---|---|---|
| Smoothing error | Target species | $10/1\sigma$ % | 0.2/0.5 |
| | Interfering species | $10/1\sigma$ % | 0.7/0.6 |
| | Retrieved parameters | | 0.6/0.6 |
| Model parameter error | Spectroscopy | 4.0/- % | 4.1/- |
| | SZA | 0.03/0.03° | 0.1/0.1 |
| | Curvature | 0.1/0.1 % | 17.2/17.2 |
| | Temperature | 1.5/2.0 K | 2.7/3.9 |
| | Zshift | 0.15/0.15 % | 2.9/2.9 |
| Measurement error | | $-/\frac{1}{\text{SNR}}$ | -/1.0 |
| Total | | | 18.2/18.1 |
| Std | | | -/15.3 |

## 3 Results and discussions

### 3.1 FTIR $C_3H_8$ retrievals at Xianghe

Figure 4 shows the *a priori* profile and retrieved profiles of $C_3H_8$. The vertical profile of $C_3H_8$ from the WACCM model shows that the $C_3H_8$ mole fraction is high near the surface and decreases with increasing altitude. Such a vertical shape is expected as
the $C_3H_8$ emissions are at the surface, and its atmospheric lifetime is too short to achieve a well-mixed troposphere. Although we perform a profile retrieval on $C_3H_8$, we only have about 1 DOF. In addition, the Tikhonov regularization matrix constrains the vertical shape when the DOF is typically close to 1.0. As a result, the retrieved $C_3H_8$ profiles have a very similar vertical shape as the *a priori* profile. However, the FTIR measurements show that the a priori column overestimates the $C_3H_8$ column concentration by about 100%. The column averaging kernel indicates the sensitivity of the retrieved $C_3H_8$ column to the $C_3H_8$
partial column in each height. Figure 4 shows that the retrieved $C_3H_8$ column has good sensitivity to all the layers, and slightly varies with SZA.

    The time series and seasonal variation of FTIR $C_3H_8$ column measurements are presented in Figure 5. To better visualize the seasonal variation, the column measurements are fitted by a periodic function $y(t) = A_0 + \sum_{k=1}^{3}(A_{2k-1}\cos(2k\pi t) + A_{2k}\sin(2k\pi t))$, where $A_0$ is the offset, and $A_1$ to $A_6$ are the periodic amplitudes, representing the seasonal variation. The
obtained mean and std of $C_3H_8$ columns at Xianghe are $1.80\pm0.81 \times 10^{15}$ $molec./cm^2$. The $C_3H_8$ columns show a high





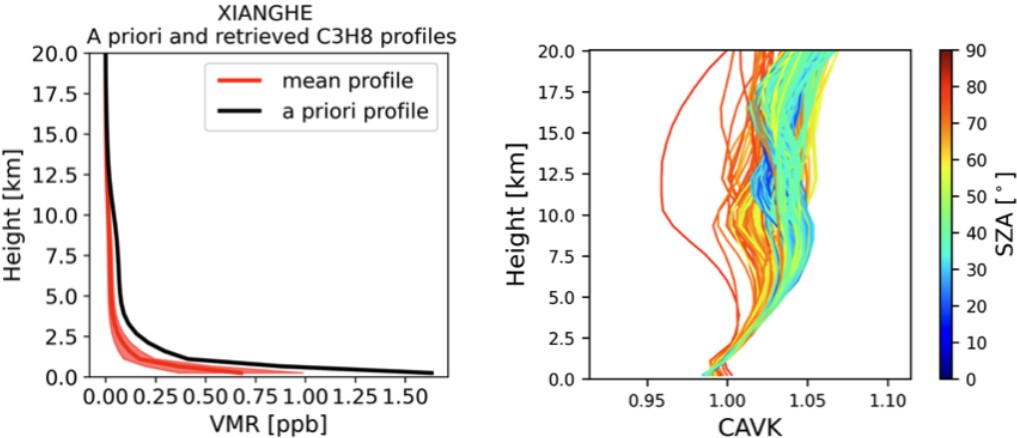

**Figure 4.** The *a priori* and retrieved $C_3H_8$ profiles (left), and the column averaging kernel (CAVK) varying with SZA (right).

mean value in July and a low value in October. The difference between the median values in July (maximum) and October (minimum) is $1.2 \times 10^{15}$ $molec./cm^2$. Although the median values of $C_3H_8$ columns in June-August are larger than those in October-March, we notice that extremely high $C_3H_8$ columns often occur in the latter period.

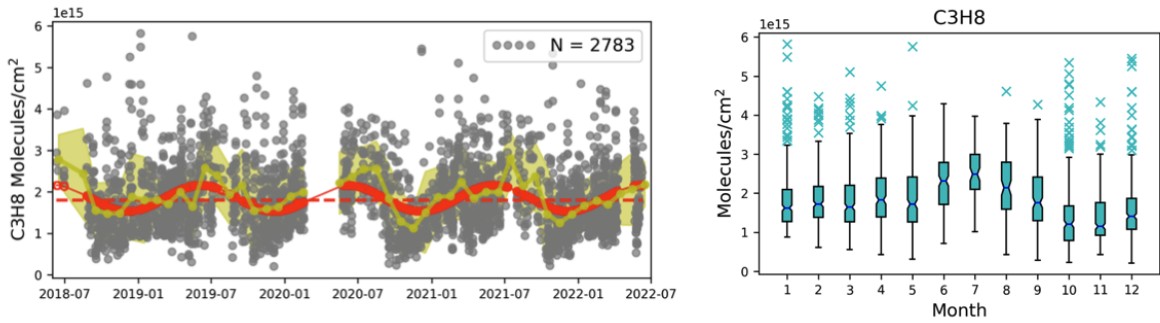

**Figure 5.** Left panel shows the time series of FTIR individual $C_3H_8$ column measurements (grey dots), monthly means (yellow line), monthly stds (yellow shade), periodic function fitting (red solid line) and the fitted offset (red dashed line). Right panel: the monthly box plot of the $C_3H_8$ columns. The bottom and top bars represent the 10% and 90% percentiles of the datasets and the blue crosses are the extremely high values above 90%.

## 3.2 FTIR $C_3H_8$ measurements against model simulations

In this section, we compare the FTIR $C_3H_8$ measurements at Xianghe with two well-known global atmospheric chemistry transport models: WACCM and Copernicus Atmosphere Monitoring Service (CAMS). The WACCM model has been widely used to generate *a priori* profiles in the NDACC-IRWG community, spanning an altitude range from the Earth's surface to the





thermosphere (Marsh et al., 2013; Gettelman et al., 2019). The horizontal resolution of the WACCM is 0.95 ° × 1.25°, with 70 vertical levels from the surface to 120 km. More information about the WACCM v6 model can be found in https://www2. acom.ucar.edu/gcm/waccm. The CAMS model (EAC4) is the fourth generation ECMWF global reanalysis of atmospheric composition, which combines model data with observations across the world. The horizontal resolution of the CAMS is 0.75 °

5  × 0.75°, with 60 model levels from the surface to ∼0.1 hPa. For more information about the CAMS model, we refer to Inness et al. (2019).

Figure 6 shows the monthly $C_3H_8$ column distributions derived from FTIR measurements, the WACCM model, and the CAMS model at Xianghe between June 2018 and December 2022. The mean and std of $C_3H_8$ column derived from CAMS and WACCM are $3.07\pm1.37 \times 10^{15} \ molec./cm^2$, and $3.00\pm1.08 \times 10^{15} \ molec./cm^2$, respectively. The mean $C_3H_8$ columns

from CAMS and WACCM models are similar, but both models are about 68% larger than the FTIR measurement. The mean difference between the model and FTIR $C_3H_8$ column is larger than the systematic uncertainty of the FTIR retrieval (∼18%). Moreover, the seasonal variations of $C_3H_8$ columns derived from the CAMS and WACCM models are different from the one derived from the FTIR measurements. CAMS and WACCM both show a low $C_3H_8$ column in summer, when the FTIR measurements present the maximum median $C_3H_8$ column. The seasonal variation of $C_3H_8$ at JPL (34°N) derived from the

ground-based MKIV spectrometer also observes a high value in summer (Toon et al., 2021), which is similar to the $C_3H_8$ seasonal variation derived from the FTIR measurements at Xianghe (39°N). Such a difference in seasonal variation between the FTIR measurements and model simulations might due to the uncertainty of emissions, transports, chemical reactions, and sinks.

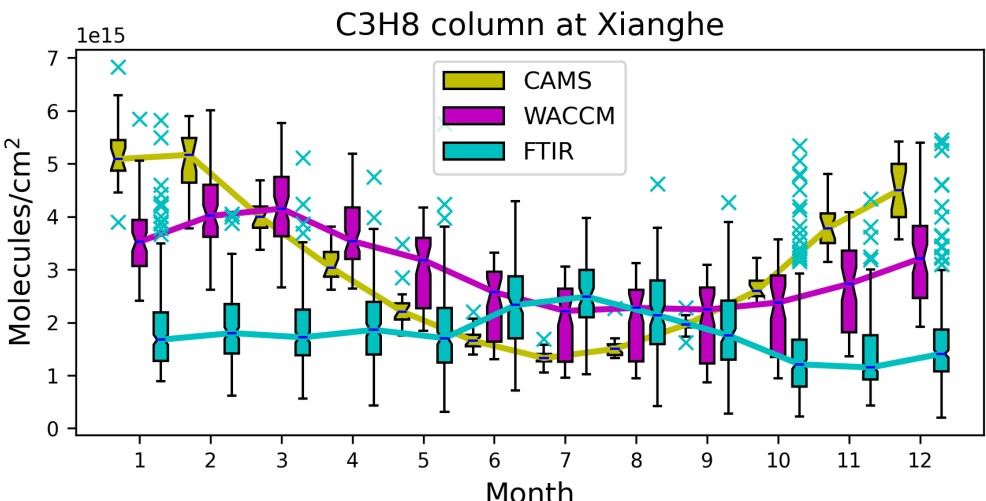

**Figure 6.** The monthly box plot of the $C_3H_8$ columns derived from the CAMS model, the WACCM model, and the FTIR measurements at Xianghe between June 2018 and December 2021.





### 3.3 Correlations with CO, CH$_4$, C$_2$H$_2$ and C$_2$H$_6$ at Xianghe

As mentioned above, the infrared spectra observed by the Xianghe FTIR system have been also used to retrieve CO, CH$_4$, C$_2$H$_2$ and C$_2$H$_6$ columns using NDACC-IRWG recommended retrieval recipes (Ji et al., 2020; Zhou et al., 2023), which allows us to investigate the correlation between C$_3$H$_8$ and these species. We are particularly interested in the correlation on a regional scale. Therefore, to reduce the impact from the background, we calculate the $\Delta$gas ($\Delta$gas = gas - monthly median) for all these species. Figure 7 shows the correlation scatter plots between $\Delta$C$_3$H$_8$ and $\Delta$CH$_4$, $\Delta$CO, $\Delta$C$_2$H$_2$, and $\Delta$C$_2$H$_6$. High correlation coefficients (R) are found between $\Delta$C$_3$H$_8$ and $\Delta$C$_2$H$_6$ (R=0.84), and between $\Delta$C$_3$H$_8$ and $\Delta$C$_2$H$_2$ (R=0.79). It indicates that the C$_2$H$_2$, C$_2$H$_6$ and C$_3$H$_8$ (NMHCs) are co-emitted in this region. The slope of $\Delta$C$_2$H$_6$ and $\Delta$C$_3$H$_8$ is 6.03$\pm$0.03, which suggests a corresponding mixing ratio of C$_2$H$_6$ and C$_3$H$_8$ mole fractions during the production in North China. CO, as a pollutant tracer, also has a good correlation with C$_3$H$_8$ (R=0.72). According to the MEIC inventory, both CO and NMHC are emitted from the energy production, industry, residential and transport sectors.

The FTIR measurements show that the correlation between $\Delta$C$_3$H$_8$ and $\Delta$CH$_4$ is relatively weak (R=0.45). Note that the variation of the CH$_4$ column is also affected by the stratospheric partial column (Sepúlveda et al., 2014). The DOF of the FTIR CH$_4$ retrieval is about 2.5 allowing us to derive the tropospheric and stratospheric CH$_4$ partial columns separately(Zhou et al., 2018). However, even after eliminating the interference from the stratosphere, the tropospheric CH$_4$ partial column still have a weak correlation with C$_3$H$_8$ (R=0.43). It is probably due to that the CH$_4$ major emissions in North China are from rice cultivation, waste, and animals instead of the oil and gas production (Ji et al., 2020), and the CH$_4$ measurements include the emissions from much farther away as compared to the C$_3$H$_8$ measurements because of its long lifetime (Callewaert et al., 2023).

To further investigate the ratio of $\Delta$C$_2$H$_6$ to $\Delta$C$_3$H$_8$, the time series of their ratios, together with the monthly correlation coefficients between both time series between June 2018 and June 2022 are illustrated in Figure 8. The ratio of each month is derived from the linear fitting using all co-located $\Delta$C$_2$H$_6$ and $\Delta$C$_3$H$_8$ hourly measurements in that month. A relatively low correlation between these two species is found in summer as compared to other three seasons. The mean and std of the ratios are 5.4$\pm$2.1 for the whole period. The ratio is lowest in summer and highest in winter, with seasonal means of 6.6, 3.8, 5.4, and 8.3 in spring, summer, autumn, and winter, respectively.

### 3.4 FTIR measurements at Xianghe against MkIV measurements

Here, the C$_3$H$_8$ and C$_2$H$_6$ columns derived from the FTIR measurements at Xianghe are compared to the ground-based MKIV C$_3$H$_8$ retrievals at 6 sites in Sweden and the USA (Figure 9). Note that the C$_3$H$_8$ and C$_2$H$_6$ retrievals from the MkIV spectrometers at 12 sites have been discussed in Toon et al. (2021), we only select 6 sites as the measurements are very limited at other 6 sites. The locations and measurement time coverages of sites used in this study are listed in Table 4.

Figure 9 shows that the C$_2$H$_6$ column is the largest at Xianghe, apart from several extremely high values at JPL-B and FTS. The seasonal variations of C$_2$H$_6$ columns are similar at these sites, especially for JPL-B, MTB and Xianghe, with a high value in northern spring and a low value in northern autumn. Note that, it is hard to derive the seasonal variation of C$_2$H$_6$ columns



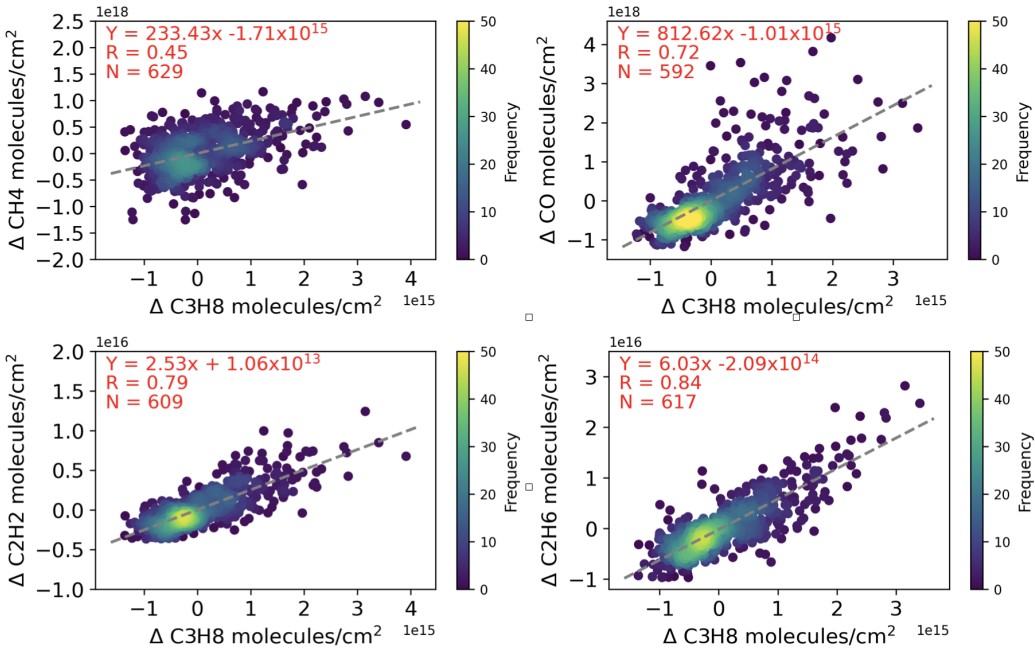

**Figure 7.** The correlation plots between co-located $\Delta C_3H_8$ and $\Delta CH_4$, $\Delta CO$, $\Delta C_2H_2$ and $\Delta C_2H_6$ hourly means at Xianghe between June 2018 and July 2022. The grey dashed line is the linear fit, N is the number of the FTIR measurements, R is the Pearson correlation coefficient.

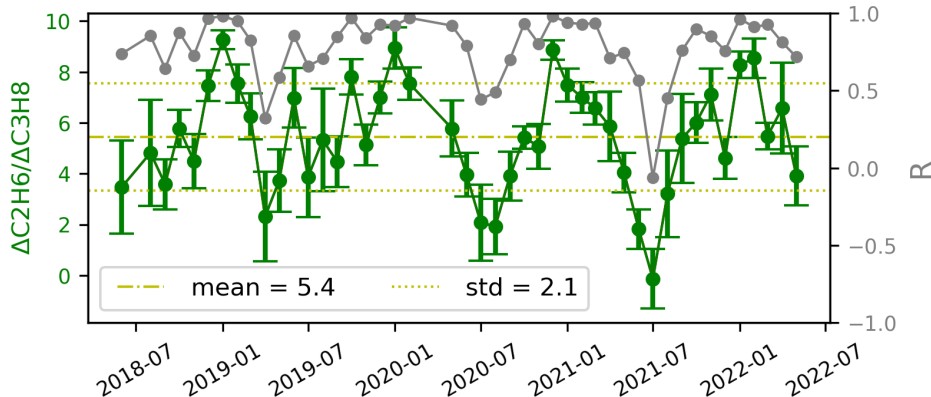

**Figure 8.** The time series of the ratio of $\Delta C_2H_6$ to $\Delta C_3H_8$ monthly means and stds (green, on the left-hand vertical axis scale), together with their monthly correlation coefficients (grey, on the right-hand vertical axis scale) between June 2018 and June 2022.

at ESN, FAI, TMF and FTS, because measurements were carried out in several months. The mean and std of $C_2H_6$ columns at JPL-B are $1.96\pm0.52 \times 10^{16}\ molec./cm^2$, which is about 25% less than that at Xianghe ((Xianghe-JPL)/Xianghe $\times 100\%$). Keep in mind that the $C_3H_8$ columns at MTB and Xianghe have been multiplied by 10 in Figure 9. The $C_3H_8$ column at





Xianghe is quite low as compared to other sites, which is only larger than that at MTB (mountain site), but much less than those at the mid-latitude sites. The mean and std of $C_3H_8$ columns at JPL-B are $2.14\pm1.33 \times 10^{16} \; molec./cm^2$, which is about 12 times larger than that at Xianghe. The seasonal variations of $C_3H_8$ columns are similar at JPL-B and Xianghe too, with a high value in northern summer and a low value in northern winter. The good correlations (R>0.6) between $C_3H_8$ and $C_2H_6$

5 columns at JPL-B and FTS have been demonstrated in Toon et al. (2021), which is similar to what we observe at Xianghe. However, the ratio of $\Delta C_2H_6$ to $\Delta C_3H_8$ at JPL-B and FTS are $0.16\pm0.10$ and $0.78\pm0.10$, respectively, which are much less than the ratio observed at Xianghe of $6.03\pm0.03$. It indicates that the emission of $C_3H_8$ is much larger in the Los Angeles basin, California than that in North China.

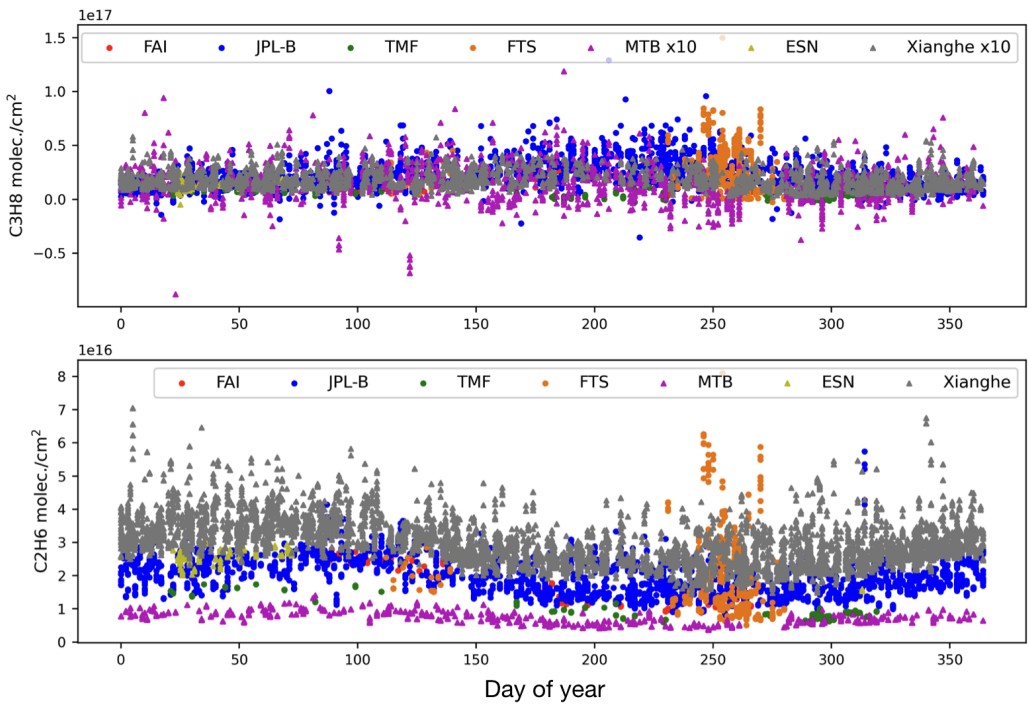

**Figure 9.** The $C_3H_8$ (upper panel) and $C_2H_6$ (lower panel) columns observed by ground-based Bruker IFS 125HR at Xianghe and MkIV spectrometer at 6 sites. Note that the $C_3H_8$ columns observed at MTB and Xianghe are multiplied by 10 to have a better view.

## 4 Conclusions

10 The Xianghe FTIR 125HR system measures the solar absorption spectra following the NDACC-IRWG guidance. For the first time, the FTIR MIR spectra at Xianghe are used for the $C_3H_8$ column retrieval, using the well-established SFIT4 code, between June 2018 and July 2022. In this study, the retrieval strategy, retrieval uncertainty, and retrieval information are presented and discussed. Due to the wide and weak absorption of $C_3H_8$, we only derive the $C_3H_8$ column instead of its vertical profile. The



**Table 4.** The locations, and data time coverages of the MkIV measurements at 6 sites, together with their mean $C_3H_8$ and $C_2H_6$ columns. The bottom row is the Xianghe FTIR measurements in this study.

| Site | Country | Latitude | Longitude | Altitude (km) | Time coverage | $C_3H_8$ (molec./cm$^2$) | $C_2H_6$ (molec./cm$^2$) |
|---|---|---|---|---|---|---|---|
| Esrange(ESN) | Sweden | 67.89°N | 21.08°E | 0.271 | Nov 1999 - Mar 2020 | $1.4 \times 10^{16}$ | $2.6 \times 10^{16}$ |
| Fairbanks(FAI) | USA | 64.83°N | 147.61°W | 0.182 | Mar-Sep 1997 | $1.4 \times 10^{16}$ | $1.8 \times 10^{16}$ |
| Mt. Barcroft(MTB) | USA | 37.58°N | 118.23°W | 3.801 | Oct 1998 - Aug 2002 | $1.4 \times 10^{15}$ | $7.3 \times 10^{15}$ |
| Ft. Sumner(FTS) | USA | 34.48°N | 104.22°W | 1.260 | Oct 1989 - Sep 2021 | $2.6 \times 10^{16}$ | $1.9 \times 10^{16}$ |
| TMF, Wrightwood (TMF) | USA | 34.38°N | 117.68°W | 2.257 | Jul-Sep 1988; Nov 1996 Jan-Aug 1998; Oct 2009 | $2.7 \times 10^{15}$ | $8.3 \times 10^{15}$ |
| JPL B183(JPL-B) | USA | 34.20°N | 118.17°W | 0.345 | Jun 1985 - Jan 2022 | $2.1 \times 10^{16}$ | $2.0 \times 10^{16}$ |
| Xianghe | China | 39.75°N | 116.96°E | 0.036 | Jun 2018 - Jul 2022 | $1.8 \times 10^{15}$ | $3.0 \times 10^{16}$ |

systematic and random uncertainties of the $C_3H_8$ retrieved column are estimated to be 18.2% and 18.1%, respectively. In the $C_3H_8$ retrieval window, $CH_4$ and $H_2O$ absorption lines are not perfectly fitted, indicating there is still room left to improving the line lists of these two species.

The mean and std of the $C_3H_8$ column derived from the FTIR measurements at Xianghe are $1.80 \pm 0.81 \times 10^{15}\ molec./cm^2$. A month-to-month variation is observed with a high value in July and a low value in October. The difference between the median values in July (maximum) and October (minimum) is $1.2 \times 10^{15}\ molec./cm^2$. The FTIR $C_3H_8$ column retrievals are compared to two well-known models (CAMS and WACCM). It is found that the mean $C_3H_8$ columns from the two models are 68% larger than the FTIR measurements at Xianghe, which is beyond the systematic uncertainty of our FTIR retrieval. Moreover, the seasonal variations of the $C_3H_8$ column derived from CAMS and WACCM models also deviates from that derived from the FTIR measurements. Further investigations are needed to better understand the mismatch between the model simulations and FTIR measurements, and to improve the $C_3H_8$ model simulations at Xianghe.

As $C_3H_8$ are co-emitted with $CH_4$, CO, $C_2H_2$, and $C_2H_6$ during oil and gas production, we calculate the correlation between $\Delta C_3H_8$ and these species at Xianghe. Good correlations are found between $C_3H_8$ and $C_2H_6$, between $C_3H_8$ and $C_2H_2$, as well as between $C_3H_8$ and CO. However, the correlation between $C_3H_8$ and $CH_4$ is relatively weak, which is probably due to $CH_4$ emission in North China being dominated by rice, cultivation, and waste, instead of oil and gas production and fossil fuels combustion. By comparing the $C_3H_8$ and $C_2H_6$ columns at Xianghe with 6 other sites around the world, provided by the ground-based MkIV spectrometers, we find that the $C_2H_6$ column at Xianghe is the largest. However, the $C_3H_8$ column at Xianghe is only larger than those observed at the mountain sites and polar sites, and it is much less than the $C_3H_8$ columns observed at mid-latitude sites in the USA.

In summary, we successfully retrieve $C_3H_8$ columns from the FTIR MIR spectra at Xianghe, which provides us with a new dataset to understand the variation of $C_3H_8$ in North China. The retrieval strategy of $C_3H_8$ in this study should work



at other Bruker 125HR FTIR sites as well, especially for those close to a city or oil and gas field, e.g., Paris, Toronto, and Boulder. Nevertheless, efforts are still needed within the NDACC-IWRG community to generate a global harmonized FTIR $C_3H_8$ column dataset.

*Data availability.* The ground-based MkIV $C_3H_8$ and $C_2H_6$ retrievals are publicly available via https://mark4sun.jpl.nasa.gov/ground.html
5 (last access date: 27 September 2022). The FTIR $C_3H_8$ retrievals at Xianghe are available upon request. The CAMS model data are publicly available via https://ads.atmosphere.copernicus.eu/ (last access date: 27 March 2024). The WACCM model data are publicly available via https://www.acom.ucar.edu/waccm/download.shtml (last access date: 27 March 2024).

*Competing interests.* The authors declare that they have no conflict of interest.

*Acknowledgements.* The author would like to thank the NDACC community for supporting the SFIT4 retrieval algorithm. We would ac-
10 knowledge all Xianghe site stuffs, Nicolas Kumps (BIRA-IASB) for the FTIR instrument maintenance. This study is supported by the National key research and development program (2023YFC3705202).

*Author contributions.* PW and MZ design the study and wrote the manuscript. MZ, BL, BD, MDM investigated the SFIT4 retrieval strategy. WN, CH and QC operate the FTIR measurements at Xianghe. GT provides the MKIV measurements. All authors have read and commented the manuscript.





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
