# Peer review of "Atmospheric propane ( $C_3H_8$ ) column retrievals from ground-based FTIR observations at Xianghe, China"

_Atmospheric Measurement Techniques, 2024_

## Author Response (AR1)

**Reply to comments raised by Referee #1.**

The original comments are in plain texts, and our replies are in **bold texts**.

This study investigates the C3H8 retrieval from ground-based FTIR spectra at Xianghe, and discuss the C3H8 column variation in North China, based on these new FTIR measurements. The technical details and uncertainty discussion are generally well provided in current form, but the result part, such as data comparison and trend explanation, are somewhat less satisfactory. Overall, I suggest the publication on AMT after presenting more information for data interpretation. Specific suggestions are listed below.

**First of all, we would like to thank you for the comments and suggestions.**

1. Method 2.3: Line 20-25: It is still not clear why perform a profile retrieval for H2O column concentration. Because each species could have large variability in vertical scale. Moreover, suggest providing more technical details about the how to perform a profile retrieval.

**More information are added now.**

**Original text: "To reduce the impact of uncertainties about the abundances of these species, these column abundances are retrieved along with the target gas mole fractions; only for $H_2O$ we perform a profile retrieval, because of its large variability."**

**Revised text:" To reduce the impact of uncertainties about the abundances of these species, CH4, O3 and HDO columns are retrieved along with the target gas mole fractions. For these three species, their profile shapes are fixed and only scaling factors are retrieved simultaneously. As H2O absorption lines are relatively strong (Table 1) and H2O variability is large in the atmosphere, we perform a profile retrieval for H2O. Therefore, the state vector includes CH4, O3 and HDO columns, as well as 47-layers' C3H8 and H2O mole fractions."**

2. Section 3.2: Since the large difference exists for seasonal variation of C3H8 column concentration between model and FTIR measurements, it would be better not present this comparison in the main text, unless the authors could provide more evidence or information to explain these differences. For example, the authors could collect some surface observation of C3H8 concentration in Xianghe or surrounding regions that used for comparison to FTIR retrieval near the surface.

**Thanks for the suggestion. Unfortunately, there is no surface observation of C3H8 in Xianghe or surrounding regions. Currently we do not have solid conclusion to fully understand the discrepency between the FTIR measurements and the model simulations. In the revised version, this part has been removed.**

3. Section 3.3 Line 20-25: What is the significance by providing the ratio of ΔC2H6 to ΔC3H8? What does the trend of this ratio mean?

**Since C2H6 and C3H8 are co-emitted by oil gas sources (Li et al., 2017; Bourtsoukidis et al., 2019) and C2H6 and C3H8 have similar lifetimes with about 2-8 weeks, the ratio of ΔC2H6 to ΔC3H8 can represent the emission ratios of C2H6 to C3H8 in this region.**

**The trend of this ratio reprents the trend of the emission ratio in this region. As our FTIR measurements do not show a clear trend in ΔC2H6/ΔC3H8, it is inferred that the emission ratios of C2H6 to C3H8 in this region remain unchanged between 2018 and 2022.**

4. Section 3.4: The authors compare FTIR measurement to MkIV data here, but the basic information about MkIV measurement were not well described. Readers might be very interested about the principle of technique used for C3H8 measurement in MkIV and the accuracy of these data. Based on these information, we can rule out the systematic difference deviation between FTIR and MkIV.

**Thanks for the suggestion. More information about the MKIV C3H8 data are added in the revised version.**
**"MKIV C3H8 data uses the GFIT inverse retrieval code to derive the C3H8 columns from the MKIV observed spectra between 2964.5 and 2970 cm$^{-1}$ with a specral resolution of 0.5 cm$^{-1}$. The mean uncertainties of the MKIV retrieved C3H8 and C2H6 column are estimated to be around $8\times10^{15}$ molecules/cm2 and $7\times10^{14}$ molecules/cm2, respectively, which are also provided by Toon et al., (2021)."**

**References:**

Bourtsoukidis, E., Ernle, L., Crowley, J. N., Lelieveld, J., Paris, J.-D., Pozzer, A., Walter, D., and Williams, J.: Non-methane hydrocarbon (C$_2$–C$_8$) sources and sinks around the Arabian Peninsula, Atmos. Chem. Phys., 19, 7209–7232, https://doi.org/10.5194/acp-19-7209- 2019, 2019.

Li, M., Liu, H., Geng, G., Hong, C., Liu, F., Song, Y., Tong, D., Zheng, B., Cui, H., Man, H., Zhang, Q., and He, K.: Anthropogenic emission inventories in China: a review, Natl. Sci. Rev., 4, 834–866, https://doi.org/10.1093/nsr/nwx150, 2017.

Toon, G. C., Blavier, J.-F. L., Sung, K., and Yu, K.: Spectrometric measurements of atmospheric propane (C3H8), Atmos. Chem. Phys., 21, 10727–10743, https://doi.org/10.5194/acp-21-10727-2021, 2021.

**Reply to comments raised by Referee #2.**

The original comments are in plain texts, and our replies are in **bold texts**.

General comments:

This study used ground-based FTRI Mid-infrared observations at Xianghe to retrieve C3H8 column through optimal estimation approach. Compared with CH4 and H2O, the absorption of C3H8 of is weak at 2964.5-2970.0 cm-1, thus retrieving C3H8 is challenging. Although the authors conducted uncertainty analysis, I still have some concerns about the accuracy and importance of the C3H8 retrievals.

 **First of all, we would like to thank you for the comments and suggestions.**

Specific comments:

1.  Due to the weak absorption of C3H8, I suspect that the a posteriori C3H8 strongly depends on a priori C3H8. Figure 6 shows the seasonal variation of FTIR C3H8 is different from the two model simulations. How is the monthly variation of the a priori C3H8? Is it similar to the FTIR retrieval? It would be interesting to see to how the a posteriori C3H8 vary if different a priori profile (for example, using profiles from CAMS and WACCM as a priori) is used.

**Thanks for the suggestion and comment. In this study, the a priori profile of C3H8 is derived from the mean of WACCM model simulations between 1980 and 2040. The a priori mole fraction profile is fixed, which does not vary with time. By using a fixed a priori profile, the retrieved C3H8 mole fraction is less affected by the a priori variation. More information is added in the revised version. In addition, we added the a priori C3H8 columns in the left panel of Figure 5. Note that the a priori mole fraction is fixed, while the a priori columns vary with surface pressure, wich a maximum in winter and minimum in summer.**

[Figure]

**Figure 5.** Left panel shows the time series of FTIR a priori columns (black dots), individual C3H8 column measurements (grey dots), monthly means (yellow line), monthly stds (yellow shade), and periodic function fitting (red solid line) and the fitted offset (red dashed line). Right panel: the monthly box plot of the FTIR retrieved C3H8 columns. The bottom and top bars represent the 10% and 90% percentiles of the datasets and the blue crosses are the extremely high values above 90%.

2. As CH4 and H2O have stronger absorption than C3H8, and CH4 and H2O absorption lines are not perfectly fitted, how it will affect the accuracy of C3H8 retrieval should be discussed.

**Thanks for the comment. More discussion about CH4 and H2O impacts are added in the revised version. Particularly, the retrieval error of the C3H8 column from the CH4 and H2O spectroscopy uncertainty are re-calculated and added in Table 3.**

3. In Sect. 2.4, how uncertainty is calculated? Apparently, we cannot get $x_t$ and $b_t$. More details are needed.

**Thanks for the comment. More information about the uncertainties are added in the revised version.**

**It is assumed that 10% of the a priori profile is used to derive the diagonal values of the systematic covariance matrix $S_a^{sys}{}_{ii} = \sigma_i^2$ , and the off-diagonal values of $S_a^{sys}$ are calculated as $S_a^{sys}{}_{ij} = \sigma_i\sigma_j$ (von Clarmann 2014 AMT). The covariance matrix derived from the WACCM 61-years' monthly means are set to the random covariance matrix $S_a^{ran}$.**

**Regarding the model parameter uncertainties in the Table 3, the systematic/random $S_b$ matrix was created by the mean/standard deviation of the differences between NCEP and ERA5 at Xianghe. The random deviation is about 2K, and the systematic deviation is about 1.5K for the whole vertical range. For the target spectroscopic parameters, the relative uncertainties of C3H8 is set to 4% according to the PLL database. For the CH4 and H2O spectroscopy parameters, the relative uncertainty of 5% is derived from the HITRAN 2020 dataset. For the uncertainties of curvature, zshift, and SZA, we use the values provided by the SFIT4 algorithm (https://wiki.ucar.edu/display/sfit4/SFIT4+Version+1.0.xx+Release), which is recommended by the NDACC-IRWG community.**

Toon, G. C., Blavier, J.-F. L., Sung, K., and Yu, K.: Spectrometric measurements of atmospheric propane (C3H8), Atmos. Chem. Phys., 21, 10727–10743, https://doi.org/10.5194/acp-21-10727-2021, 2021. should be cited. It used observation around 2967 cm$^{-1}$ to retrieve C3H8.

References:

*von Clarmann, T.: Smoothing error pitfalls, Atmos. Meas. Tech., 7, 3023-3034, doi:10.5194/amt-7-3023-2014, 2014.*

**Reply to comments raised by Referee #3.**

The original comments are in plain texts, and our replies are in **bold texts**.

Review of "Atmospheric propane (C3H8) column retrievals from ground-based FTIR observations at Xianghe, China" by Zhou et al., amt-2024-67

This manuscript presents a 4 year data record of C3H8 (and C2H6) column amount measurements at the Xianghe TCCON site. Retrievals are carried out from high resolution spectra in the 2964.5-2970.0 cm-1 range (C3H8 Q-branch). Several spectroscopic line lists are compared in terms of spectral fit quality. The uncertainty of retrieved C3H8 columns is estimated and the seasonal variation, as well as the correlation of C3H8 with other hydrocarbons, is discussed. Results are compared to two different models and to FTIR measurements at other sites.

The subject of this paper has relevance within the scope of AMT. New data are presented and I find the overall quality of presentation clear and concise.

Yet, the paper currently leaves the reader with a number of open questions. Since Toon et al. (2021) have already presented first measurements of C3H8 with a similar methodology, I find that more work is needed with respect to the interpretation of the results before publication, so that this paper adds value to the existing literature. The big question to me is: Do we trust these column retrievals and if so, why are C3H8 columns so low in Xianghe?

**First of all, we would like to thank you for the comments and suggestions.**

General comments:

G1: I am concerned that the retrieved C3H8 values from this study do not line up with similar measurements and models. I do find it surprising that C3H8 columns 50 km downwind of Beijing should be one order of magnitude lower than C3H8 columns in Pasadena (10^15 vs. 10^16 molec/cm2). You mention on page 2: "we expect that the C2H6 and C3H8 concentrations are relatively high in this region" - I agree and would like to ask you to please expand your study in a way that resolves this apparent conflict. Similarly, it appears that the present measurements are in broad agreement with CAMS/WACCM during wildfire season in northern summer, but not during the winter when gas is being consumed. What are the atmospheric situations when you do observe high wintertime C3H8? Can we understand these results in the context of the existing measurements and models?

Thanks for the comments. We expect that the C2H6 and C3H8 concentrations are relatively high in this region, which is consistent with the model simulations (both CAMS and WACCM models). For example, the C3H8 mole fraction in the CAMS model (Figure 1A) shows that Beijing region is a hotspot around the world, which is larger than Pasadena. When comparing the Xianghe FTIR measurements with MKIV measurements at JPL, the C3H8 columns are much smaller than that at JPL. Two aspects may cause this. First, the uncertainty of the emission inventories used in the model is very large. Pétron et al., (2014) pointed out that the uncertainty in the C3H8 emission estimate is larger than 30%, which is the sum of the relative uncertainty in the total CH4 emission estimate and the relative uncertainty in the CH4-to-C3H8 slope. What's more, the uncertainty of the emission can be much larger in a small region as compared to a national level. Secondly, the retrieval uncertainty of the MKIV measurements is also very large. Toon et al., (2021) reported the uncertainty of MKIV C3H8 measurement is about 8.0x10^15 molecules/cm2. More information has been added in discussion of the revised version.

The discrepancy in the seasonal variation between the FTIR measurements at Xianghe and model simulations (CAMS and WACCM) is also found at JPL, where the model shows a maximum in winter and a minimum in summer but the MKIV measurements also show a maximum in summer and a minimum in winter. First, we checked the sources and sinks in the model and tried to understand why the model shows a maximum in winter and a minimum in summer. The emission dataset used in the CAMS model is shown in Figure A2. The seasonal variation of C3H8 emission is relatively small, with an amplitude of less than 15%. In addition, we also checked the OH seasonal variation in the CAMS model (Figure A3). The model shows that OH is high in summer, which is about 2 times larger than that in winter. Therefore, combining the sources and sinks in the model, the C3H8 concentration in the model is larger in winter as compared to that in summer. However, as we motioned above, the uncertainty of the C3H8 emission is relatively large (>30%), and the uncertainty of the OH in the model is also quite large. All these uncertainties may lead to a different seasonal variation of C3H8. Secondly, we checked the FTIR C3H8 fitting residual and temporal variation of the FTIR measurements to understand whether there is an artificial in its seasonal variation or not. Figure A4 shows that there is no clear difference in the fitting residual between summer and winter. We also took a month in summer and winter, separately (Figure A5), and checked the temporal variation of C3H8 and C2H6 simultaneously. In both months, the FTIR measurements show that C3H8 columns have a high correlation with C2H6 columns. The day-to-day temporal variation observed by FTIR C2H6 measurements can also be well captured by FTIR

**C3H8 measurements. Therefore, we believe the FTIR C3H8 measurements are reliable in both seasons.**

**In summary, the discrepancy in the seasonal variation between the FTIR measurements at Xianghe and model simulations is probably caused by the model uncertainty. However, we have no solid conclusion. Following the comment and suggestion proposed by Referee #1, we have removed the section about the comparison between the FTIR measurements and model simulations in the revised version for now.**

[Figure]

**Figure A1. The C3H8 mass fraction near the surface in June 2020 from the CAMS model.**

**CAMS–GLOB–ANT Anthro propane v6.2 monthly**
monthly (2018–2024)

[Figure]

**Figure A2. The time series of the C3H8 monthly emissions in Asia and North America from the CAMS-GLOB-ANT database.**

**Figure A3. The seasonal variation of OH concentration near the surface at Xianghe from the CAMS model.**

[Figure]

**Figure A4. The time series of the monthly mean and SD of the fitting RMSEs.**

[Figure]

**Figure A5. The time series of daily mean and SD of FTIR C3H8 and C2H6 measurements at Xianghe in January (upper panel) and June (bottom panel) 2019.**

G2: Please discuss why the residual structure in Fig. 2 has structures that are larger than the ones found by Toon et al. (2021). Please present average spectral fit residuals in Fig. 2 (with std/min/max) to build confidence that your fit works. Are the average fit residuals understood?

**Thanks for the comment. The fit residuals in Fig.2 is from only 1 spectra (the spectra shown in Fig. 1), while the Fig.2 in Toon et al., (2021) shows the average of 5000 MKIV spectra. In the revised version, we added the mean and standard deviation of the fitting residuals from all 2783 C3H8 retrievals at Xianghe. The mean RMS is 0.317%, which is slightly better than 0.3658% reported in Toon et al., (2021). In addition, the RMSE in Table 3 has been corrected. The original values in Table 3 are the values given by the SFIT4 code, which is RMSE/mean(y) but not RMSE. Mean(y) is the mean of the transmittance in this window. Now, we have recalculated the RMSE.**

[Figure]

**Figure A6. The mean (black line) and SD (grey shadow) of the fitting residual from 2783 FTIR C3H8 retrievals at Xianghe between June 2018 and July 2022.**

Minor comments:

M1: You note the range of spectral resolution you have across the full MIR range. What is the spectral resolution in your window at ~2970 cm-1?

**The spectral resolution is 0.0051 cm$^{-1}$.  Added.**

M2: Page 4, lines 21-22: add C2H6 to list of interfering gases

**Done.**

M3: the matrix S (not S_R) on page 6, line 16 and in eq. 8 needs to be introduced in the text

**Added.**

M4: You calculate the retrieval error due to uncertainties in spectroscopic parameters, but did not include all relevant parameters like line position or line shift. Please explain why these can be neglected in the uncertainty budget.

**Thanks for the comment. In fact, we calculated all the model parameters, i.c. wavenumber shift, solar line intensity, solar line wavenumber shift, max optical path difference, instrument line shape, as the uncertainties derived from these parameters are less than 0.1%, they are not listed in Table 3.**

M5: How do you fit the spectral baseline (aka "background curvature")? Please explain in the text.

**We use a linear regression (y=ax) to fit the spectral baseline. The background slope is included in the state vector (*x*). Since the retrieval window is relatively small of about 5 cm$^{-1}$, we do not apply the second order fitting on the spectral background. Therefore, the background curvature is included in the model parameter vector (*b*) but not retrieved.**

M6: Page 11, lines 14-19 and page 14, lines 14-16: Please provide a more convincing explanation why the C3H8-CH4 correlation is weaker than the correlation with other NMHCs. There are oil and gas producing/consuming regions in Northern China (e.g. Changqing oil field/city of Beijing) and with a life time on the scale of weeks-months natural gas related C3H8 emissions from such basins/cities could easily reach Xianghe (compare page 2, lines 20-23).

**Thanks for the comment. On one hand, according to the inventory, C3H8 and C2H6 are mainly emitted by oil and gas product, which is also a source of CH4. However, CH4 has a multiply sources, and the major CH4 emissions in North China are rice cultivation, waste, and animals instead of the oil and gas production. On the other band, CH4 is a much longer lifetime as compared to C3H8 and C2H6, therefore, CH4 measurements includes the signal from a larger**

**region as compared to C3H8 and C2H6 measurements. As a result, C3H8-CH4 correlation is weaker than the correlation with other NMHCs.**

M7: How long is the integration time for one measurement?

**One measurement takes about 10 minutes.**

M8: I do not understand Fig. 9: why are there negative values for C3H8?

**Thanks for the comment. The negative values of C3H8 are from the MKIV retrievals based on the spectral fitting provided by Toon et al., (2021). As the negative values has no physical meaning, we have filtered them out in the revised version.**

M9: Is it possible that the C3H8 regularization is too loose?

**As the DOF of C3H8 is already close to 1.0 and the retrieved C3H8 vertical profile shape is smilar to that of the a priori vertical profile shape, we believe that the C3H8 regularization is not loose. This is also true because of the weak absorption of C3H8. It is not possible to get too much vertical information from the observed spectra.**

M10: If differences between retrievals and a priori data are big: How trustworthy are the a priori profiles of C3H8 from WACCM and are they very different from the ones in CAMS? Would it not make sense to scale only the lower layers of the C3H8 profile, instead of fitting/scaling the full profile, especially since the retrieved profiles do not appear to differ substantially from the prior above ~10 km? How many layers are there below 10 km?

**The a priori columns are added in the Fig.5. We use the average WACCM model simulations between 1980 and 2040 as the a priori mole fraction profile, which does not vary with time. This fixed a priori profile is recommended by the NDACC-IRWG community. By doing this, we can reduce the impact from the prior.**

**There are 47 vertical layers between the surface and the top of the atmosphere (120 km) included in the SFIT4 code at Xianghe, with 15 layers below 10 km. The retrieved profiles also change above 10km. As the C3H8 mole fraction is very low above 10km, it is not so visible from the figure.**

M11: Which interfering species has the most impact on C3H8 retrieval accuracy/precision?

**H2O and CH4 have relatively large impacts on C3H8 retrieval. We have added the uncertainty estimation from these two interfering species in Table 3.**

M12: Have you checked the correlation between C3H8 and H2O/HDO? What did you find?

**See Figure A7. The correlation coefficient value between C3H8 and H2O columns is 0.5. In general, their correlation is not so significant.**

[Figure]

**Figure A7. The scatter plots between daily C3H8 and H2O columns at Xianghe.**

M13: Maybe mention somewhere that the DLR and HITRAN2020 line lists for H2O are very similar; I believe for the main isotopologue: HITRAN2020 = DLR, except for line positions, but better to double-check.

**Thanks for the comment. We have double-checked the line parameters in the HITRAN2020 and DLR in this window. They are similar, but still slightly different in both line position and line intensity. Table A1 shows an example of all the H2O main isotope (11) lines between 2965.5 and 2966.5 cm-1 in these two line lists.**

**Table A1. All the H2O main isotope (11) lines between 2966.0 and 2966.5 cm-1 in these two line lists.**

| DLR | | HITRAN2020 | |
|---|---|---|---|
| wavenumber | Line intensity | wavenumber | Line intensity |
| 2966.006188 | 1.63E-22 | 2966.006245 | 1.628E-22 |
| 2966.128478 | 6.88E-25 | 2966.068863 | 8.539E-30 |
| | | 2966.457227 | 1.240E-30 |
| | | 2966.480092 | 3.721E-30 |

Technical comments:

T1: page 2, line 4: "of the order of 10 years" -> "on the order of 10 years"

T2: caption of Table 1: "specie" -> "species"

T3: page 6, line 16 and caption of Table 3: 2040 -> 2004

T4: page 6, eq. 8: S_ij -> S_{ij}

T5: page 7, line 13: pressure-dependent parameter, temperature-dependent parameter -> pressure-dependence parameter, temperature-dependence parameter

T6: page 10, line 17: "might due" -> "might be due"

T7: page 11, line 14: "separately(" -> separately ("

T8: page 11, line 16: "have" -> "has"

T9: page 11, line 16: "it is probably due to that the" -> "it is probably due to the fact that the"

**Thanks. All the technique comments has been corrected.**

**References:**
**Pétron, G., et al. (2014), A new look at methane and nonmethane hydrocarbon emissions from oil and natural gas operations in the Colorado Denver-Julesburg Basin, J. Geophys. Res. Atmos., 119, 6836–6852, doi:10.1002/2013JD021272.**

[revised manuscript text omitted]